# Spatial patterns and environmental influences of COVID-19 outbreaks, post-Omicron

Aleksandra Stamper[1,2]*, Rachel E. Baker[1,2]

**1** Department of Epidemiology, School of Public Health, Brown University, Providence, Rhode Island, United States of America, **2** Institute at Brown for Environment and Society, Brown University, Providence, Rhode Island, United States of America

* aleksandra_stamper@brown.edu

## Abstract

The seasonality of many respiratory pathogen outbreaks, such as influenza and respiratory syncytial virus, is driven by climate factors, such as specific humidity or temperature. However, it remains unclear whether climate plays a role in determining the seasonality of COVID-19, given that the evolution of novel strains likely plays a key role in shaping outbreak dynamics. Here we use Emergency Department data to explore spatial differences in COVID-19 outbreak dynamics over three years, from April 2022 through March 2025. We observe that outbreak patterns varied across latitude, with southern states experiencing larger summer peaks and northern states facing more evenly distributed summer to winter outbreaks or larger winter peaks. We find that specific humidity and temperature at the state level are significantly associated with observed differences in ED visits with a COVID-19 diagnosis, even after controlling for state-level variation in vaccination status. Our results imply a role for climate in influencing COVID-19 outbreak dynamics. We anticipate these findings will provide a foundational understanding of factors shaping SARS-CoV-2 transmission as COVID-19 becomes endemic in the United States.

## Introduction

COVID-19, a respiratory disease caused by severe acute respiratory syndrome coronavirus 2 (SARS-CoV-2), was declared a global pandemic by the World Health Organization (WHO) on March 11, 2020. The pandemic has resulted in significant global disease burden, with an estimated 1.2 million deaths in the United States alone as of September 2025 [1]. Early regression analyses suggested a potential relationship between climate factors (e.g., temperature, humidity, precipitation, and solar radiation) and SARS-CoV-2 transmission [2–4]. However, evaluating these relationships in the early stages of a pandemic is challenging due to the large susceptible population, meaning preliminary findings may fail to fully capture longer-term patterns or underlying climate-disease dynamics [5]. As COVID-19 shifts from a global pandemic into an

**Data availability statement:** Aggregated COVID-19 and climate data are publicly available. Code and data to run the analysis is available at https://github.com/aleksandrastamper/covid-seasonality. R was used for the statistical computing environment (version 4.4.3) to process, analyze, and visualize data.

**Funding:** A.S. and R.E.B. are supported by the Burroughs Wellcome Fund award number 1181130 https://www.bwfund.org. The funders had no role in study design, data collection and analysis, decision to publish, or preparation of the manuscript.

**Competing interests:** The authors have declared that no competing interests exist.

endemic disease, the shrinking susceptible population enables more robust analysis of climate influences driving SARS-CoV-2 transmission [5,6].

Many respiratory viruses exhibit seasonal outbreak trends linked to cyclical shifts in climate factors such as temperature or specific humidity [7,8]. Influenza [9,10] or respiratory syncytial virus [RSV] [11] commonly experience winter peaks in temperate regions, driven by troughs in specific humidity [12]. Other viruses, such as enterovirus, are known to peak during the summer in temperate climates, corresponding with higher temperatures [13]. When outbreak patterns differ along a latitudinal gradient, this may indicate a role for climate in influencing disease seasonality, as changes in latitude broadly map onto differences in weather patterns [14]. For many common respiratory viruses, such as influenza [15], coronaviruses [16], Human metapneumovirus [17], or RSV [18], the viral structure contains an outer lipid membrane (i.e., an envelope) surrounding the viral genome. Climate factors such as temperature [19,20], humidity [19,21,22], and UV radiation [23,24] can disrupt the stability of the lipid bilayer, making enveloped viruses more susceptible to environmental conditions compared to their non-enveloped counterparts. Given the established roles of climate drivers in other respiratory viruses such as influenza [9] or RSV [11], there has been strong interest in whether climate influences may be linked to COVID-19 outbreak patterns.

In the early stages of the pandemic, numerous studies assessed links between climate and COVID-19 transmission. Several analyses detected associations between colder, drier conditions and elevated disease activity [2,25–30], while others identified more complex, nonlinear relationships [20,31]. However, the large susceptible population during this period likely minimized the influence of climate on transmission dynamics [5]. As susceptibility wanes and COVID-19 transitioned toward endemic circulation, climate factors may have played a large role on outbreak dynamics, enabling their effects on transmission to be inferred more clearly. Following the Omicron wave in the United States, COVID-19 activity settled into a recurring cycle of summer and winter outbreaks (S1 Fig), exhibiting a biannual outbreak pattern that diverges from other common respiratory diseases with annual outbreaks (e.g., influenza [19], RSV [11], or enterovirus [13]). The United States, spanning temperate and subtropical climates across latitudes [32], presents an ideal setting to further examine the climate-disease relationship, particularly given the presence of publicly available high-resolution COVID-19 data starting in 2020 [33].

Here, we perform a state-level analysis in the United States to assess the associations between climate factors and COVID-19 trends, using weekly emergency department (ED) data on the percentage of visits with a COVID-19 diagnosis. The study period ranges from April 1, 2022, through March 30, 2025. We used the percentage of ED visits with a COVID-19 diagnosis rather than positive test counts, as it offers a more stable measure of disease burden and is less affected by reporting or testing behaviors. To characterize COVID-19 activity, we calculate two standardized disease metrics (epidemic intensity and mean case timing) to state latitude values (derived from centroid coordinates of each state). Epidemic intensity (scaled 0–1) reflects the concentration of cases over a year, and mean case timing represents the weighted

average week in which the cases occur. Finally, we use fixed effects linear regression analyses that account for state-level vaccination coverage to explore the relationships between specific humidity, temperature, and COVID-19 activity.

## Materials and methods

### Data

ERA5 hourly reanalysis data for 2m temperature and dew point temperature were downloaded at 0.25° × 0.25° resolution (roughly 30 km$^2$ grids) [34], and daily values were extracted for each state using state-level shapefiles [35]. Specific humidity and relative humidity were computed from temperature and dew point temperature using standard Magnus formulations for actual and saturation vapor pressure, which are approximations of the Clausius-Clapeyron relation [36]. The joint distributions of specific humidity and temperature were compiled in S2 fig. Centroids for each state (representing the geographic midpoint of their latitude and longitude coordinates) were calculated using the $st_{centroid()}$ function from the *sf* package [37,38]. Data on the weekly percentage of ED visits with a COVID-19 diagnosis at the state level were downloaded from the CDC COVID Data Tracker [33]. Data on COVID-19 vaccination information at the state level were downloaded from the CDC COVID-19 Vaccination Trends data catalog [39]. Alaska and Hawaii were excluded to focus analyses on contiguous US states. A complete-case analysis was conducted, excluding Wyoming in 2024–25 due to discontinuation of state-level reporting.

### Outbreak characteristics

Epidemic intensity (EI) is defined based on Dalziel et al. [40] as:

$$EI = 1 - \frac{\sum p \ln(p)}{\ln\left(\frac{1}{52}\right)} \tag{1}$$

Where $p$ is a vector of the mean percentage of ED visits with a COVID-19 diagnosis per week divided by the sum across all weeks. EI was calculated for average weekly activity over the entire time frame and for each "COVID-19 year", i.e., April 2022 – March 2023, April 2023 – March 2024, and April 2024 – March 2025 (S3 fig).

Mean case timing (MT) was calculated by identifying the center of gravity with the "circular" package in R [41]. The calculation identifies the arithmetic weighted mean week of ED visits with a COVID-19 diagnosis. The formula is given by:

$$MT = 52 \cdot \text{atan2} \frac{\left(\sum_{w=1}^{52} I_w \sin(w), \ \sum_{w=1}^{52} I_w \cos(w)\right)}{2\pi} \tag{2}$$

Where $w$ is the week of the year (ranging 1-52) converted to radian units by $w = \frac{week}{52} \cdot 2\pi$, and $I_w$ is the mean percentage of ED visits with a COVID-19 diagnosis in week $w$, averaged across April 2022 – March 2025.

The ratio of Winter to Summer peak size was calculated for each COVID-year per state. Weeks were designated as Summer (April – September) or Winter (October – March), and the maximum peak size was extracted from each season before calculating the ratio of the winter peak divided by the summer peak.

### Fixed effects binned regression

We ran a binned fixed effects regression, with temperature bins of 5 °C and specific humidity bins of 2 g/kg to estimate outcome log(ED visits with a COVID-19 diagnosis) as:

$$\begin{aligned} \log(y_{it}) = {} & \beta_1 t_{[-20,-15)} + \beta_2 t_{[-15,-10)} + \cdots + \beta_9 t_{[25,30)} \\ & + \gamma_1 q_{[0,2)} + \gamma_2 q_{[2,4)} + \cdots + \gamma_6 q_{[18,20)} \\ & + \alpha_i + \alpha_y + \alpha_w + \varepsilon_{it} \end{aligned} \quad (3)$$

Where $log(y_{it})$ is the logged percentage of ED visits with a COVID-19 diagnosis in state $i$ at week and year $w$; $\beta$ terms represent dummy variables corresponding to the temperature bins; $\gamma$ represents dummy variables corresponding to the specific humidity bins; $\alpha_i$ refers to state-specific fixed effects; $\alpha_y$ refers to year-specific fixed effects; $\alpha_w$ refers to week-specific fixed effects, and $\varepsilon_{it}$ reflects clustering of standard errors at the state level. A binned fixed effects model with only specific humidity as a predictor had a lower adjusted $R^2$ value compared to the full model (0.6126 vs. 0.6120 S7 fig) . When implementing the model, intercept bins were selected to align with the troughs in Fig 2C, at 10-12 g/kg specific humidity and 15-20°C temperature. Multiple bin sizes were tested for each parameter (ranging 2-5°C for temperature and 2-4 g/kg for specific humidity), and the bin combination resulting in the lowest AIC (Akaike Information Criterion) and BIC (Bayesian Information Criterion) values was selected for the final model SI tbl2.

## Generalized additive model

To investigate the relationship between climate and COVID-19 activity, we ran a generalized additive model (GAM) with outcome log(ED visits with a COVID-19 diagnosis) in a particular week, as:

$$\log(y)_i \sim s(q_i, k = 5) + t_i + s(week, bs = \text{``cc''}, k = 20) + factor(state), \quad i = 1, \dots, n \quad (4)$$

Where $log(y)_i$ logged ED visits with a COVID-19 diagnosis in week $i$, specific humidity in week $i$ was included as a smooth term estimated with a thin plate regression spline with 5 knots ($q_i$), and temperature in week $i$ was included as a linear term ($t_i$). The model additionally includes a cyclic smooth term for the week of year to capture seasonal patterns and state fixed effects to control for time-invariant differences across states. The GAM used thin plate regression splines ($bs = \text{``tp''}$), the default spline basis in the R $mgcv$ package [42–46]. The specific humidity smooth used $k = 5$ basis functions to capture the observed U-shaped association. Knot placement was determined internally by $mgcg$'s thin-plate basis construction, which minimizes the integrated squared second derivative. Smoothing parameters were estimated using restricted maximum likelihood (REML), which provides stable smoothing parameter estimates and reduces the risk of undersmoothing. We report the effective degrees of freedom (edf), $X^2$, and p-values for each term in SI tbl3. Model residuals were examined, and we included plots describing residual vs. fitted values, residuals vs. linear predictors, the Q-Q plot, and a histogram of residuals in the supplement (Supplemental S5 fig) in the supplement. The final GAM was selected by maximizing the adjusted $R^2$ and percentage deviance explained (SI tbl4). Based on results from the binned fixed effects regression analysis, a spline term was chosen to reflect the U-shaped relationship between specific humidity and cases (S6 fig), while a linear term was selected to represent the relationship between temperature and cases.

## Fixed effects model with vaccination

To isolate the effect of climate on COVID-19 cases, we ran additional fixed effects models that incorporate vaccination status at the state level:

$$\log(y)_{l,t} = \beta_1 C_{l,t} + \beta_2 V_{l,t} + \gamma_l + \lambda_t + \varepsilon_{l,t} \quad (5)$$

Where $log(y)_{l,t}$ is logged ED visits with a COVID-19 diagnosis in week $t$ and state $l$; $C_{l,t}$ refers to the climate variables (mean specific humidity, mean temperature) at time $t$ and state $l$; $V_{l,t}$ represents the percentage of the population with a completed COVID-19 booster at time $t$ and state $l$; $\gamma_l$ represents the state-specific fixed effects, and $\lambda_t$ represents the

year-specific fixed effects. Separate regressions were run for each climate factor. Standard errors were clustered at the state level.

### Sensitivity analyses

Multiple sensitivity analyses were conducted to assess the robustness of the climate-COVID-19 relationship including testing alternative outcome definitions, model specifications, and addressing potential sources of temporal and spatial dependence.

**Alternative outcome definition (COVID-19 ED visits versus COVID/Influenza-like-illness [ILI] ratio).** As the percentage of ED visits with a COVID-19 diagnosis can vary with changes in overall ED volumes, we re-estimated the GAM using an alternative outcome: the log ratio of ED visits with a COVID-19 diagnosis to ILI visits. Weekly state-level ILI visits were obtained from CDC FluView [47]. The GAM in Eq 4 was fit to both outcomes, and found that the estimated smooth terms for specific humidity and week remained highly consistent, indicating that the climate associations are not driven by shifting denominator features in the ED percentage metric (SI tbl1).

**Serial correlation-robust inference.** The weekly structure of the outcome data may introduce heteroskedasticity and autocorrelation in the GAM residuals. To address this, we re-estimated the GAM using Newey-West heteroskedasticity- and autocorrelation-consistent (HAC) standard errors [48]. We computed HAC-adjusted standard errors and performed joint Wald tests for the specific humidity and week-of-the-year smooth terms (SI tbl3).

**Spatial autocorrelation on residuals.** To evaluate whether residuals exhibited spatial dependence across states, we computed Moran's I, a measure of spatial autocorrelation, using queen-contiguity (border or corner adjacency) spatial weights with 999 permutation simulations. Analyses were conducted on state-level mean residuals for the primary GAM.

**Assessing multiple climate variables.** To evaluate whether the results depended on the choice of climate variables, we re-fit the GAM using: a) relative humidity in place of specific humidity, b) dew point temperature in place of air temperature, and c) combinations of these variables while maintaining the same model structure and fixed effects (SI tbl4).

### Results

We observe a latitudinal gradient in COVID-19 outbreak patterns (e.g., epidemic intensity and mean case timing) at the state level, as represented in Fig 1. In Fig 1a, we plot the scaled average weekly percentage of ED visits with a COVID-19 diagnosis on the state level between April 2, 2022, and March 29, 2025. The cyclical nature of the plot highlights the year-round COVID-19 infection trends, indicating the relative summer and winter peak sizes in each state. While most states exhibited two periods of heightened activity per year, corresponding with summer and winter months, southern states (e.g., Florida, Georgia, or Texas) tended to have larger summer peaks. In contrast, more northern states (e.g., Maine, Ohio, or Rhode Island) experienced summer and winter peaks of more comparable magnitude or larger winter peaks. Large summertime outbreaks contributed to higher epidemic intensity and earlier mean case timing in southern states compared with their more northern counterparts. Higher-latitude states exhibited larger winter outbreaks relative to summer activity, with states above 39 degrees latitude showing average winter-to-summer outbreak ratios greater than 1 S4 fig. Fig 1B shows a strong gradient in epidemic intensity and mean case timing in the United States, with linear regression confirming significant relationships where higher latitude was associated with a reduction in epidemic intensity ($p <<< 0.001$) and a later mean case timing ($p <<< 0.001$). A year-by-year analysis of epidemic intensity showed that the latitude–EI relationship was largely robust to annual variability. Lower latitude remained significantly associated with higher outbreak intensity in the 2022–23 and 2023–24 seasons ($p <<< 0.001$; S3 fig). In contrast, during the 2024–25 season, a substantial winter outbreak that disproportionately affected higher-latitude states attenuated this relationship, yet the association between latitude and epidemic intensity was not statistically significant ($p > 0.05$).

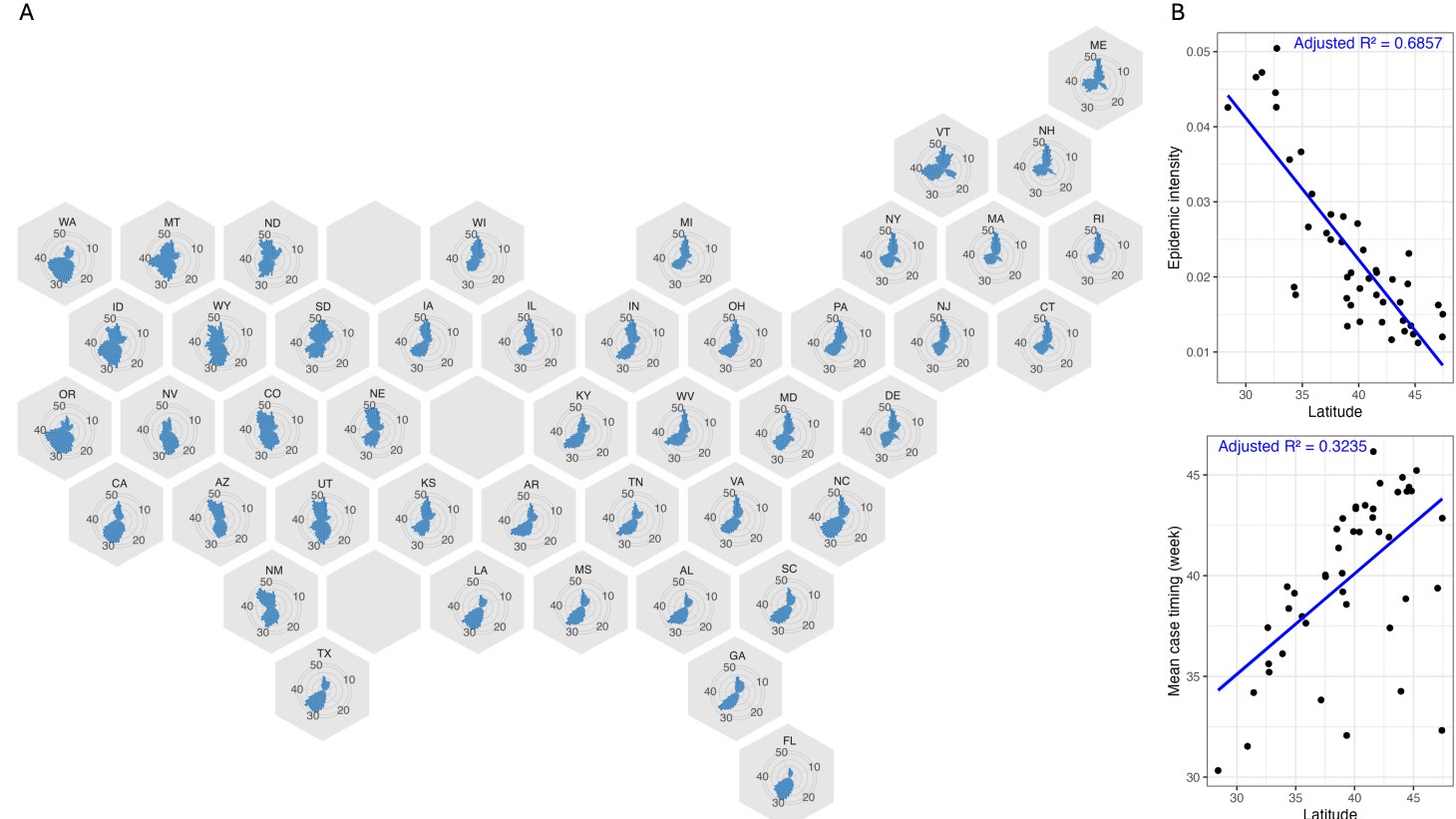

**Fig 1**. **Assessing COVID-19 infection patterns through the percentage of weekly emergency department visits with a COVID-19 diagnosis, April 2022–March 2025.** Note: Wyoming data represents April 2022 – March 2024 due to halting data reporting. **A.** Scaled mean weekly percentage of ED visits with a COVID-19 diagnosis during study period per state. **B.** Scatterplots showing association between latitude and mean timing of cases (week) and epidemic intensity across 2022-2025.

Heatmaps showing scaled weekly ED visits with a COVID-19 diagnosis (Fig 2A) highlight the persistence of biannual activity across latitudes, with consistent summer and winter peaks in higher-latitude states and more dominant summer peaks at lower latitudes. Next, we explore whether average outbreak patterns differ across states grouped by mean temperature and mean specific humidity. When examining weekly disease burden by temperature and specific humidity groupings, clear regional differences emerge: southern states with hotter, more humid climates experienced the largest outbreaks in the summer, whereas the states with colder, less humid climates exhibited summer and winter peaks of more comparable size (Fig 2B). The larger summer peaks in the lower-latitude states drive the higher epidemic intensity values seen in Fig 1B, compared to the more evenly dispersed biannual peak sizes observed in the more northern states. Generalized additive model (GAM) fits stratified by mean state-level specific humidity further illustrate how outbreak characteristics varied across climates (Fig 2C). Specific humidity exhibited a marginal U-shaped association with disease activity, with elevated ED visits with a COVID-19 diagnosis occurring at both high and low levels, while intermediate values were associated with reduced disease burden. Temperature also showed a marginal non-linear relationship with disease activity, with higher case burden at colder and warmer temperatures, and lower disease burden at intermediate temperature ranges. Notably, the U-shaped relationships between COVID-19 activity and climate factors become more pronounced in

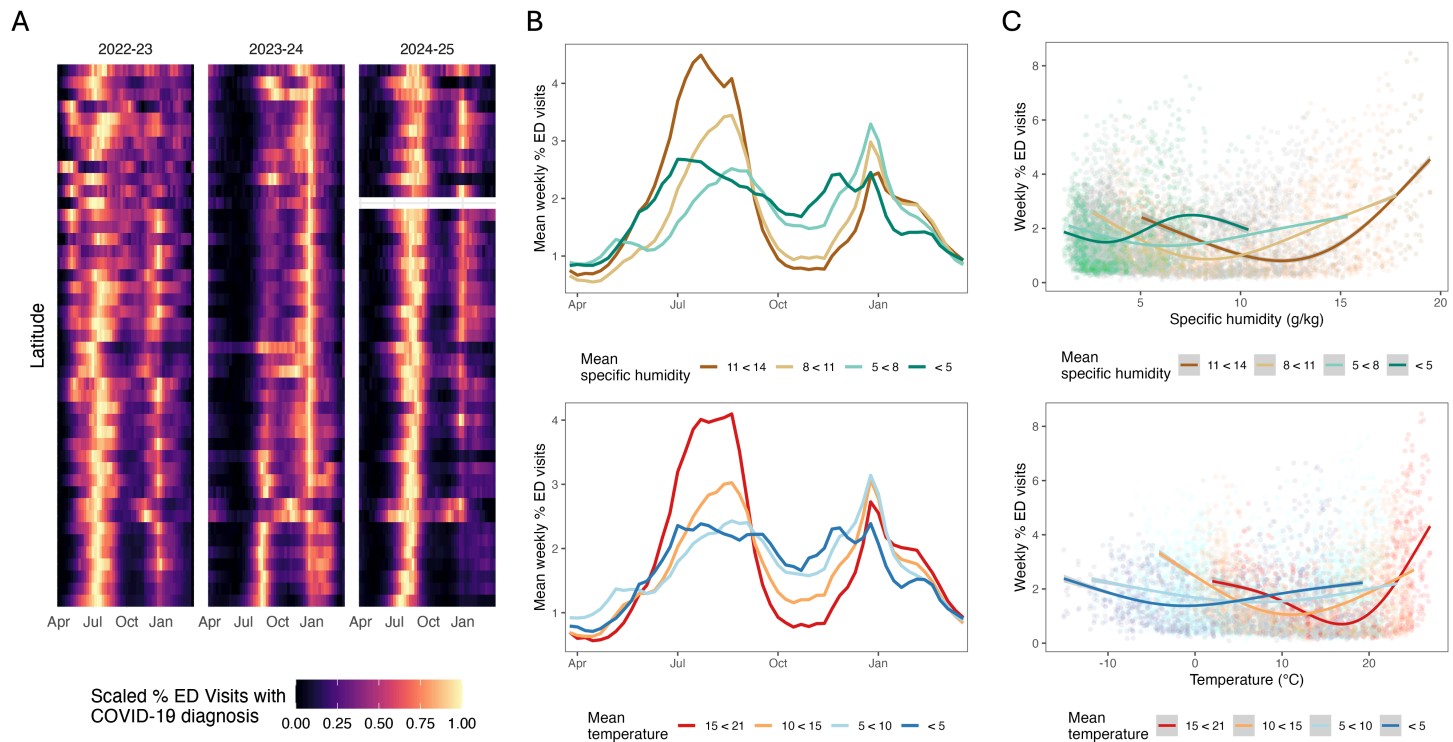

**Fig 2.** **COVID-19 outbreak patterns following a latitudinal gradient, which may be related to an identified U-shaped relationship with climate factors. A.** Heat maps of scaled weekly COVID-19 burden per "COVID-year" (April - March), ordered by descending latitude. **B.** Averaged weekly percentage of ED visits with a COVID-19 diagnosis over the COVID-year, by climate subgroup. **C.** Relationship between climate factors (specific humidity, temperature) and COVID-19 activity showing a U-shaped relationship between weather and case burden, with lower-latitude locations exhibiting more extreme case troughs along the climate range.

southern states, with sharper increases in disease burden associated with higher temperatures and specific humidity values. This suggests that while climate-COVID-19 associations are broadly consistent across the United States, outbreaks in lower latitude states may be more strongly influenced by climate factors compared to higher latitude states.

To further investigate the potential climate-disease relationship, we next quantified the joint effects of temperature and specific humidity on COVID-19 activity using a two-step modeling framework (Fig 3). First, binned regression was used to investigate specific humidity, temperature, and COVID-19 activity. We found a U-shaped relationship between COVID-19 burden and specific humidity, where the lowest disease risk occurred at intermediate specific humidity levels, and a more monotonic, negative association with temperature (Fig 3A). To test whether this identified relationship, particularly the apparent increase in summer activity, was driven by overall changes in ED visits, we estimated a fixed effects model using the ratio of ED visits with a COVID-19 diagnosis to ILI reports. This sensitivity check yielded results consistent with the main model, confirming a significant nonlinear association between specific humidity and disease activity SI tbl1. Given our binned model implies a nonlinear effect of specific humidity and a more linear fit of temperature, we fit a generalized additive model (GAM) to weekly logged ED visits with a COVID-19 diagnosis, incorporating a spline term to represent a nonlinear U-shaped relationship for specific humidity and a linear relationship with temperature. Multiple climate variables were tested when developing the GAM; a nonlinear combination of specific humidity with a linear term for temperature explained the greatest proportion of variance (SI tbl4). Drawing from the observed ranges of temperature and specific humidity, Fig 3B shows the predicted surface of the climate-COVID-19 relationship, highlighting how the risk of disease is expected to vary under different combinations of temperature and specific humidity. Lower disease risk is

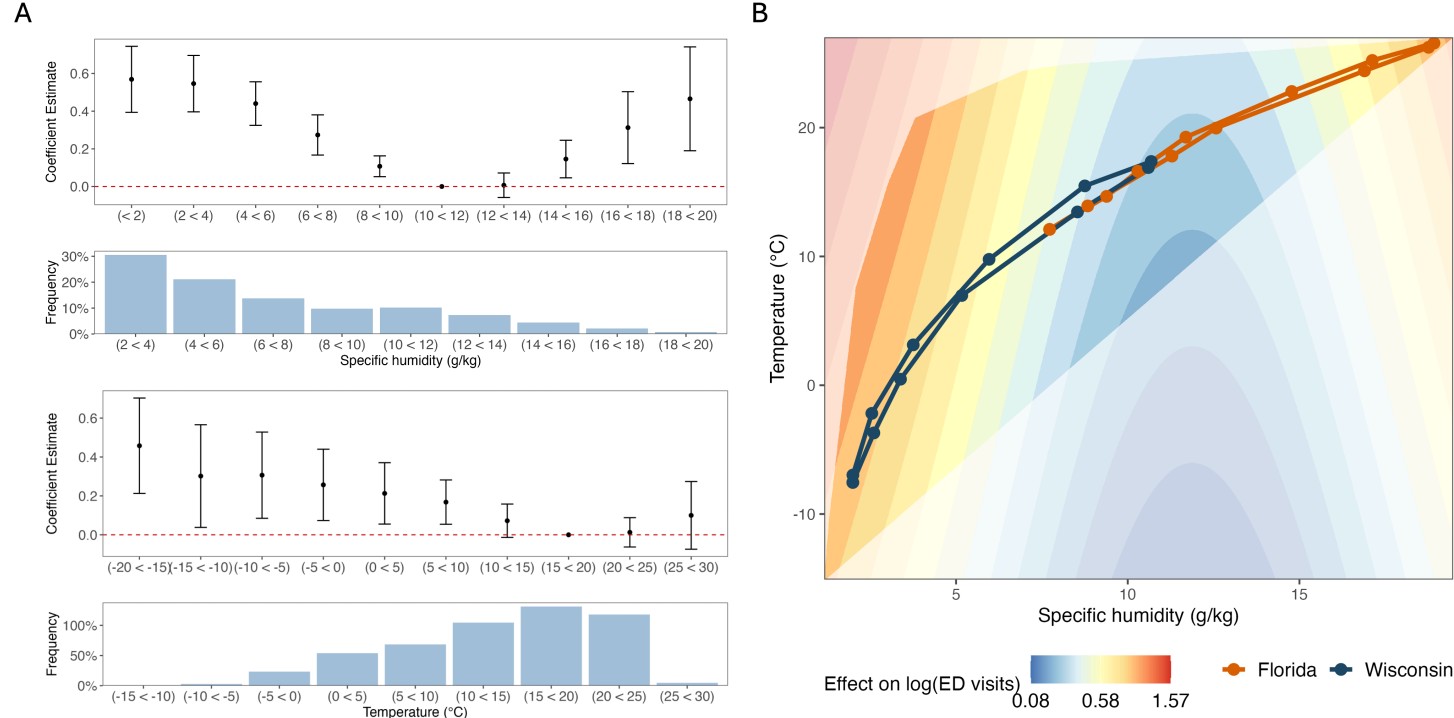

**Fig 3. Modeled relationship between climate predictors (specific humidity, temperature) and logged ED visits with a COVID-19 diagnosis.** **A.** Results from the binned fixed effects model, showing a U-shaped relationship between specific humidity and logged ED visits, and a more linear relationship between temperature and logged ED visits. **B.** Surface plot indicating climate contexts of predicted heightened COVID-19 activity from a generalized additive model whose model specification was influenced from the binned fixed effects model estimates for temperature and specific humidity. The joint distribution of temperature and specific humidity in the entire dataset is found in S3 fig, while the marginal effect of specific humidity is seen in S8 fig.

predicted at intermediate levels of temperature and specific humidity, whereas higher risk emerges at the upper range of these conditions. The more vibrant polygon in Fig 3B indicates the range of observed co-occurring temperature and specific humidity combinations during the study period. The monthly average climate combinations for Wisconsin (a northern state) and Florida (a southern state) are plotted on the GAM surface to show the trajectory of temperature and specific humidity throughout the year. The transmission surface conveys how the hot and high specific humidity combinations in Florida underlie the summer outbreak, while the cool and low specific humidity conditions in Wisconsin promote its winter outbreak, with lower anticipated COVID-19 activity occurring in the middle of the climate ranges. Together, these results suggest that climate factors may amplify COVID-19 activity, particularly at the upper or lower bounds of temperature or specific humidity.

## Discussion

Similar to other endemic respiratory diseases (e.g., influenza, RSV, enterovirus), this analysis suggests that COVID-19 outbreak patterns may be influenced by climate factors. Beyond the spatial component of latitude, specific humidity and temperature accounted for excess variation in state-level outbreak patterns not explained by latitude alone. COVID-19 activity exhibited a nonlinear relationship with specific humidity, with outbreaks concentrated in both summer and winter months, where the magnitude and seasonal distribution of these outbreaks varied along a latitudinal gradient. These findings remained consistent even when accounting for sources of spatial and temporal bias in the outcome data (SI tbl3),

testing other climate variables (SI tbl4), and accounting for the percentage of the population with a completed COVID-19 booster dose (SI tbl5). States with higher mean specific humidity and temperature generally exhibited higher epidemic intensity values due to the presence larger summer outbreaks. Understanding how variations in outbreak patterns across geography may be related to climate factors helps improve public health preparedness and strategic resource allocation.

Our analysis identified a strong, statistically significant signal in which higher and lower values of specific humidity were associated with elevated disease activity, while temperature had a more negative association. The mechanisms underlying this pattern are uncertain, and the literature shows limited consensus on the climate conditions most conducive to COVID-19 activity, particularly when focusing on the later stages of the pandemic (i.e., post-2021). As COVID-19 transitions into endemic circulation, observed biannual peaks are likely influenced by a combination of viral evolution (which depletes the size of the susceptible population), climate influences, and population factors (e.g., vaccination coverage). Early multinational analyses found a negative association between temperature and SARS-CoV-2 infections [25], and similar negative correlations were identified between temperature and COVID-19 mortality across temperate regions of Europe and the United States through mid-2021 [49]. An analysis in Japan, a country spanning temperate and tropical climates, found that COVID-19 activity had a negative correlation with temperature but a J-shaped relationship with specific humidity [50]. Laboratory-based studies also suggest nonlinear relationships: SARS-CoV-2 half-life varies with relative humidity, following a U-shaped relationship with the strongest effects seen above 22°C [20]. We find strong evidence supporting this U-shaped relationship, especially prominent in southern regions of the United States with higher average levels of specific humidity, where the size of the summer COVID-19 outbreak peak was prone to eclipse the winter peak S7 fig. Taken together, these findings indicate that the climate–COVID-19 relationship is nuanced and shaped by the joint distribution of temperature and specific humidity.

There are several caveats with this analysis. First, while the regression analysis identifies an association between climate and COVID-19 outbreak patterns, a mechanistic model is required to identify the link between climate and transmission. Such a model could be used to assess the likelihood of continued summertime outbreaks in the coming years. Second, our results rely on three years of post-Omicron data. Continued data collection at the sub-national, or ideally sub-state level, is necessary to better identify possible climate links. The discontinuation of some SARS-CoV-2 data collection efforts limits our ability to test sub-state level variation and influences on transmission. Finally, while our fixed effect model removes common state-level factors, there may be other characteristics or social factors that influence SARS-CoV-2 dynamics. Longer time series, adjustment for the the COVID-19 stringency index [51], or the incorporation of data from other countries may help improve the assessment of a climate effect. More broadly, additional work is needed to understand the current drivers of SARS-CoV-2 transmission, and how the interplay between viral evolution, vaccination, and behavioral or environmental factors may influence outbreak patterns in the coming years.

## Conclusion

Our results show distinct variation in COVID-19 outbreak patterns along a latitudinal gradient in the United States following the Omicron wave in 2022. While most states experienced two outbreaks per year in summer and winter, southern states were more likely to have a larger summer peak, whereas more northern states commonly had two similar size peaks during summer and winter. Additionally, we found that climate variables (specific humidity and temperature) were significantly associated with case activity, even after adjusting for state-level vaccination coverage, assessing an alternative outcome measure, and performing sensitivity analyses. These findings were robust to multiple years of data, even with the arrival of new SARS-CoV-2 variants during the study period. While multiple factors likely influence the seasonality of COVID-19 outbreaks, we found evidence supporting a potential role for climate in modulating transmission dynamics.

## Supporting information

**S1 Fig.Weekly timeseries displaying the percentage of ED visits with a COVID-19 diagnosis per state for the study period.** Most states had three years of complete data, while Wyoming stopped reporting in 2025 and thus only used data from April 2022 - March 2024. Each "COVID-19 year" has a different color to indicate the demarcation of the study period.
(PNG)

**S2 Fig. Joint distribution of observed temperature and specific humidity values across all states in dataset.** Warmer colors indicate more commonly observed temperature and specific humidity combinations.
(PNG)

**S3 Fig. Epidemic intensity calculated for each state and COVID-year in relation to latitude.** Color indicates year.
(PNG)

**S4 Fig. Ratio of the Winter/Summer COVID-19 peak size per state.** Winter/Summer peak ratio for each state and COVID-year, where larger ratios indicate larger summer peaks relative to winter peaks.
(PNG)

**S5 Fig. Diagnostic plots for the generalized additive model (Eq 4).** A. Q-Q plot showing residuals largely following the theoretical normal distribution. B. Histogram of residuals, approximately symmetric and centered near zero. C. Residuals versus linear predictor, showing no major nonlinear patterns, with mild heteroskedasticity. D. Observed versus fitted values, indicating that fitted values span a narrower range than the observed log(ED visits). Together, these diagnostics support the adequacy of the model while motivating inclusion of state and cyclic week effects.
(PNG)

**S6 Fig. Estimated smooth terms from the GAM including state and cyclic week effects.** A. Specific humidity shows a nonlinear, U-shaped association with log(ED visits). B. The cyclic spline for week captures within-year seasonal structure. Solid lines represent fitted effects; dashed lines show 95% confidence intervals.
(PNG)

**S7 Fig. Results from a binned fixed effects model assessing logged ED visits with a COVID-19 diagnosis with specific humidity as a predictor.** Results from the binned fixed effects model,showing a U-shaped relationship between specific humidity and logged ED visits.
(PNG)

**S8 Fig. Marginal effect of specific humidity on the predicted logged ED visits with a COVID-19 diagnosis.** Estimated marginal effects at temperatures of 0 °C, 10 °C, and 20 °C.
(PNG)

**S1 Table. Comparison of climate smooth terms across outcome definitions: percentage of COVID-19 ED visits (main model) vs. log ratio of COVID-19 ED to ILI ED (robustness model).** The direction, magnitude, and significance of humidity and seasonal smooths remain highly consistent across models, indicating robustness to denominator shifts in ED case mix.
(XLSX)

**S2 Table. Model performance across temperature and specific humidity bin sizes.** Sensitivity analyses to determine optimal temperature and specific humidity bin sizes. Temperature bins of 5°C paired with specific humidity bins of 2g/kg yielded the lowest AIC and BIC values and were selected as bin sizes for the fixed effects models.
(XLSX)

**S3 Table. Summary of smooth terms from the generalized additive model (GAM).** Smooth term estimates from the GAM, showing model flexibility (EDF), reference degrees of freedom, and significance of the specific humidity- and week-related spline components.
(XLSX)

**S4 Table. Sensitivity of GAM results to alternative humidity and temperature specifications.** Across all models, the humidity smooth term remains highly significant, and model fit metrics vary minimally, demonstrating that the main climate associations are not sensitive to the choice of humidity measure.
(XLSX)

**S5 Table. Quantifying the effects of specific humidity and temperature on COVID-19 burden, adjusting for the percentage of the population with a completed booster dose.** After adjusting for the percentage of a state's population who received a booster dose, we still find that higher weekly specific humidity is associated with heightened disease activity, along with lower temperature.
(XLSX)

## Acknowledgments

We thank the Editor and anonymous reviewers for reviewing this manuscript.

## Author contributions

**Conceptualization:** Rachel E. Baker.

**Data curation:** Aleksandra Stamper.

**Formal analysis:** Aleksandra Stamper.

**Funding acquisition:** Rachel E. Baker.

**Investigation:** Aleksandra Stamper.

**Methodology:** Rachel E. Baker.

**Project administration:** Rachel E. Baker.

**Supervision:** Rachel E. Baker.

**Visualization:** Aleksandra Stamper.

**Writing – original draft:** Aleksandra Stamper.

**Writing – review & editing:** Aleksandra Stamper, Rachel E. Baker.

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
