## [Decision Letter · Decision Letter 0]

30 Nov 2025

PONE-D-25-57696

Spatial patterns and environmental drivers of COVID-19 outbreaks, post-Omicron

PLOS ONE

Dear Dr. Stamper,

Thank you for submitting your manuscript to PLOS ONE. After careful consideration, we feel that it has merit but does not fully meet PLOS ONE’s publication criteria as it currently stands. Therefore, we invite you to submit a revised version of the manuscript that addresses the points raised during the review process.

Your manuscript was reviewed by two experts in the field. Both reviewers identified many important problems in your submission, and one reviewer provided copious comments. Please review the attached comments and respond point-by-point.

We look forward to receiving your revised manuscript.

Kind regards,

Yury E Khudyakov, PhD

Academic Editor

PLOS ONE

Journal Requirements:

“A.S. and R.E.B. are supported by the Burroughs Wellcome Fund award number 1181130.”

“A.S. and R.E.B. are supported by the Burroughs Wellcome Fund award number 1181130. https://www.bwfund.org”

“A.S. and R.E.B. are supported by the Burroughs Wellcome Fund award number 1181130. https://www.bwfund.org”

5. We note you have included a table to which you do not refer in the text of your manuscript. Please ensure that you refer to Table S1 in your text; if accepted, production will need this reference to link the reader to the Table.

6. We notice that your supplementary figures are uploaded with the file type 'Figure'. Please amend the file type to 'Supporting Information'. Please ensure that each Supporting Information file has a legend listed in the manuscript after the references list.

Reviewers' comments:

Reviewer's Responses to Questions

**Comments to the Author**

1. Is the manuscript technically sound, and do the data support the conclusions?

Reviewer #1: Yes

Reviewer #2: Partly

2. Has the statistical analysis been performed appropriately and rigorously?

Reviewer #1: I Don't Know

Reviewer #2: No

3. Have the authors made all data underlying the findings in their manuscript fully available?

Reviewer #1: Yes

Reviewer #2: Yes

4. Is the manuscript presented in an intelligible fashion and written in standard English?

Reviewer #1: Yes

Reviewer #2: Yes

5. Review Comments to the Author

Reviewer #1: The authors investigated the impact of climate patterns on COVID-19 outbreaks during the post-Omicron wave. The methods are appropriate, and the content is innovative. However, one point requires correction: the citation in Line 5 currently appears as “[?]” and should be fixed.

Reviewer #2: The manuscript analyzes U.S. state–level emergency department (ED) data (April 2022–March 2025) to describe post-Omicron COVID-19 seasonality and investigate associations with temperature and specific humidity using descriptive metrics (epidemic intensity and circular mean timing) and panel regressions/GAMs. It reports (i) a transparent latitudinal gradient in outbreak patterns and (ii) a U-shaped relationship between specific humidity and burden, with higher risk at both low and high values and lowest risk at intermediate humidity; temperature shows a more negative/monotone association overall. The analysis relies on public ED visit percentages from the CDC, ERA5 climate data, and state-level booster coverage, with figures summarizing temporal patterns and model-based response surfaces.

The topic is timely and suitable for PLOS ONE, and the dataset window (post-Omicron) is a strength. The manuscript offers a valuable post-Omicron perspective and clearly documents the geographic structure in COVID-19 ED patterns. However, several methodological decisions limit interpretability. With stronger attention to outcome denominators, serial/spatial dependence, bimodal timing, and potential confounders, in addition to fuller model transparency and robustness checks, the conclusions can be considered well-supported, and the claims about climate associations will be much more compelling and publication-ready.

Major comments

- Using the percentage of ED visits with a COVID-19 diagnosis is defensible for mitigating testing/reporting artifacts. However, it is sensitive to changes in the ED case mix (e.g., heat waves, injuries, RSV/flu surges) that alter the denominator independent of COVID-19 transmission. Please:

o Re-run key analyses with an incidence-style outcome (e.g., estimated COVID-related ED visits per 100k population, if derivable from the CDC series) and/or with the log ratio of COVID ED visits to a stable internal control (e.g., total respiratory visits).

o At a minimum, control for state-week total ED volume (or include week×region fixed effects) to absorb denominator shifts. This control is crucial for interpreting the U-shape at high humidity/temperature in southern states, where summertime non-COVID ED volume is large.

- EI is computed based on the mean weekly curve aggregated across April 2022 and March 2025. This collapses interannual variation (including variant waves) and may inflate smoothness. Please compute EI per COVID-year (Apr–Mar) by state, analyze variance across years, and show that the latitude relationship persists year by year. Sensitivity of results to this design choice should be presented (main text or Supplement).

- The paper summarizes timing with a single circular mean across years, but the distribution is explicitly biannual (summer and winter peaks). A single mean is not a sufficient statistic and can be misleading (means can land in troughs). Please report additional timing metrics: (a) circular dispersion, (b) separate peak weeks (summer/winter) per state/year, and (c) peak asymmetry (summer minus winter magnitude). Consider re-running the latitude regressions on these peak-specific outcomes.

- Equation (3) includes state, year, and week fixed effects—a good start—but more detail is needed. Please:

o Clarify whether two-way or three-way FE are included simultaneously (state FE + year FE + week-of-year FE). If week FE is common across years, it absorbs seasonality; then the climate effect is identified through deviations from the seasonal mean. Make this explicit and discuss implications.

o Justify bin choices (5°C, 3 g/kg) analytically (e.g., using information criteria or cross-validation) rather than aligning reference bins to visual troughs (risk of post-hoc selection).

o Show partial dependence plots with CIs and within-R² to emphasize within-state identification.

- For the GAM [Equation (4)], please specify the exact basis, knot placement, smoothing parameter selection (REML is noted), and present edf, χ²/F-tests, and residual diagnostics (ACF of residuals, concurvity).

- Weekly panel data will exhibit strong serial correlation. Clustering by state is necessary but not sufficient if residuals are highly persistent. Please report tests/plots of residual autocorrelation and repeat regressions with Driscoll–Kraay or Newey–West (panel) robust errors (or at least state-level AR(1) correction) to verify inference stability.

- Neighboring states share weather and mobility catchments; ignoring spatial autocorrelation can bias SEs and inflate significance. Perform Moran’s I on residuals and, if material, implement spatial HAC corrections or include region×week FE. Alternatively, estimate a spatial error model on state-level annual summaries as a sensitivity analysis.

- Summer/winter peaks in ED COVID percentages may be entangled with RSV/flu/enterovirus waves, school terms, AC/indoor time, wildfire smoke, or holiday periods. At a minimum, add controls for:

o Influenza and RSV ED indicators (or ILI/SARI proxies) by state-week.

o Mobility or policy stringency (the Discussion notes this as future work; it belongs in robustness today).

o School in/out of session indicators.

o Wildfire smoke (PM2.5) has affected recent summers, particularly those with smoke-affected conditions. Include these in sensitivity models to demonstrate that climate associations are not proxies for unmeasured seasonal behaviors.

- The booster covariate is treated as a single percentage; please specify the series (which booster definition? updated 2023/2024 formulations?), data cadence, and lag structure. Consider lagged vaccination and age-adjusted coverage, as state averages can mask significant age/uptake differences. Show that results are robust to alternative vaccination specifications and to inclusion of prior-wave burden as a proxy for immunity.

- The paper occasionally reads as if climate “drives” the dynamics of COVID-19. Given the observational design, please temper language to “is associated with” throughout the Abstract, Author Summary, Results, and Conclusion, and emphasize that climate may “modulate” transmission conditional on variant immune escape and behavior. The Discussion begins to do this, but still infers “amplify” in places. Tighten wording, please.

- Since Apr 2022 covers multiple Omicron sublineages with differing immune escape, please include period or variant-era indicators (or break the sample) and test for parameter stability (e.g., climate coefficients pre- vs post-XBB/BA.5 transitions). A figure that locates major lineage turnovers on the time axis would be helpful.

- The fixed-effects bins suggest a U-shape for humidity and a negative slope for temperature; the GAM surface then shows the highest predicted burden at the high-humidity/high-temperature corner for Florida and at cold/dry for Wisconsin. To avoid over-interpretation:

o Quantify uncertainty on the surface [e.g., contours of standard error of the mean (SEM) or bootstrap].

o Show the empirical distribution of (T, q) combinations (you plot a polygon of support; add heat-density so readers see where the model extrapolates).

o Report marginal effects of humidity at several temperature strata (and vice versa).

- Given literature heterogeneity (absolute vs. relative humidity, dew point, WBGT, humidex, specific enthalpy, UV), include a sensitivity table swapping relative humidity, dew point, and a solar/UV proxy (ERA5 clear-sky UV if available) to show that findings are not metric-dependent.

Minor comments

- Recommendations for figures:

o Fig 1: Note Wyoming data truncation explicitly in the caption.

o Fig 2: For heatmaps and grouped curves, include N(states) per subgroup and shaded uncertainty (e.g., state bootstraps).

o Fig 3: The caption has “Fig 3. Fig 3.” duplicated; fix. Overlay monthly points with error bars for Florida/Wisconsin and add a legend for the support polygon. Provide a color bar with units (log-percent ED COVID).

- Recommendations for reproducibility & data availability:

o The GitHub link and statement are helpful; please archive a frozen release (e.g., Zenodo or similar) with a DOI, include environment lockfiles (R version noted as 4.4.3; also pin package versions), and provide download scripts for the ERA5 and CDC series so that others can exactly reproduce your extraction window and state mappings.

o Clearly document which CDC endpoints were used, their as-of date, any backfill handling, and whether data revisions occurred during analysis.

- “N/A” is written in the Ethics Statement section. Because the analysis utilizes publicly available, aggregate surveillance data, add a sentence in the Methods section stating that the data are de-identified and publicly available, and that no human subjects review was required (cite the data sources).

- Ensure the funding statement appears verbatim in the manuscript Acknowledgments/Financial Disclosure as required by PLOS ONE.

- #5: Replace the “1.2 million deaths as of September 2025 [?]” citation placeholder with a verifiable source.

- Fix typos (e.g., “the the stringency index” in #191) and grammatical errors (e.g., “Data on COVID-19 vaccination data on the state level …” in #55 and #56 OR “becomes” in #128).

- In Materials and methods, clarify spatial aggregation: ERA5 0.25° grids → state-level averages (population-weighted or area-weighted?). Specify the state centroid source for latitude and whether Alaska/Hawaii were included or excluded.

- Please note missingness: e.g., Wyoming stopped reporting in 2025; describe how weeks with missing ED data are handled (listwise deletion? imputation?). Provide a completeness table by state×year.

6. PLOS authors have the option to publish the peer review history of their article (what does this mean?). If published, this will include your full peer review and any attached files.

Reviewer #1: **Yes: **Yutaro Akiyama

Reviewer #2: No

---

## [Author Response · Author response to Decision Letter 1]

19 Dec 2025

We are thankful for the reviewers for raising some additional statistical and sensitivity analyses to strengthen the robustness of the findings. In response, we conducted a suite of robustness checks, including examining alternative outcome definitions and model specifications, and addressing potential sources of temporal and spatial bias. These analyses confirmed the stability of our primary results. We added four supplemental tables and three supplemental figures summarizing sensitivity analyses, model assumption checks, and annual state-level outbreak metrics. We believe these measures sketch out a more robust and comprehensive analysis investigating the association between climate factors and COVID-19 activity following the Omicron wave. Detailed, point-by-point responses are provided below.

Reviewer #1: The authors investigated the impact of climate patterns on COVID-19 outbreaks during the post-Omicron wave. The methods are appropriate, and the content is innovative. However, one point requires correction: the citation in Line 5 currently appears as “[?]” and should be fixed.

Thank you for noting this citation did not transfer, we have corrected it to link to the WHO COVID-19 dashboard, and is listed first in the references as: “COVID-19 deaths | WHO COVID-19 dashboard;. Available from: https://data.who.int/dashboards/covid19/cases.”

Reviewer #2: The manuscript analyzes U.S. state–level emergency department (ED) data (April 2022–March 2025) to describe post-Omicron COVID-19 seasonality and investigate associations with temperature and specific humidity using descriptive metrics (epidemic intensity and circular mean timing) and panel regressions/GAMs. It reports (i) a transparent latitudinal gradient in outbreak patterns and (ii) a U-shaped relationship between specific humidity and burden, with higher risk at both low and high values and lowest risk at intermediate humidity; temperature shows a more negative/monotone association overall. The analysis relies on public ED visit percentages from the CDC, ERA5 climate data, and state-level booster coverage, with figures summarizing temporal patterns and model-based response surfaces.

The topic is timely and suitable for PLOS ONE, and the dataset window (post-Omicron) is a strength. The manuscript offers a valuable post-Omicron perspective and clearly documents the geographic structure in COVID-19 ED patterns. However, several methodological decisions limit interpretability. With stronger attention to outcome denominators, serial/spatial dependence, bimodal timing, and potential confounders, in addition to fuller model transparency and robustness checks, the conclusions can be considered well-supported, and the claims about climate associations will be much more compelling and publication-ready.

Major comments

A. Using the percentage of ED visits with a COVID-19 diagnosis is defensible for mitigating testing/reporting artifacts. However, it is sensitive to changes in the ED case mix (e.g., heat waves, injuries, RSV/flu surges) that alter the denominator independent of COVID-19 transmission. Please:

Re-run key analyses with an incidence-style outcome (e.g., estimated COVID-related ED visits per 100k population, if derivable from the CDC series) and/or with the log ratio of COVID ED visits to a stable internal control (e.g., total respiratory visits).

At a minimum, control for state-week total ED volume (or include week×region fixed effects) to absorb denominator shifts. This control is crucial for interpreting the U-shape at high humidity/temperature in southern states, where summertime non-COVID ED volume is large.

Thank you for raising this important concern. To address potential denominator instability in the percentage of ED visits with a COVID-19 diagnosis, we re-estimated our primary GAM using an alternative outcome: the log ratio of weekly COVID-19 ED visits to weekly state-level ILI visits (CDC ILINet). ILI serves as a stable respiratory control that is much less likely to be affected by fluctuations in non-respiratory ED volume. We added one to ILI counts to accommodate zero-activity weeks. The revised GAM, identical to Equation 4 except for the outcome, produced highly consistent smooth terms for specific humidity and week (NEW Supplemental Table 1). The alternative model explained slightly more deviance (44.4% vs. 37.1%). These results indicate that our findings are robust to changes in denominator behavior.

(NEW) Supplemental Table 1. Comparison of climate smooth terms across outcome definitions: percentage of COVID ED visits (main model) vs. log ratio of COVID ED to ILI ED (robustness model).

B. EI is computed based on the mean weekly curve aggregated across April 2022 and March 2025. This collapses interannual variation (including variant waves) and may inflate smoothness. Please compute EI per COVID-year (Apr–Mar) by state, analyze variance across years, and show that the latitude relationship persists year by year. Sensitivity of results to this design choice should be presented (main text or Supplement).

This is an important point, as our original EI metric aggregated data across all COVID-years. In response, we recalculated EI separately for each COVID-year (Apr–Mar) and presented these results in NEW Supplemental Figure 2. Year-specific EI–latitude associations were generally consistent with the pooled estimates (Figure 1). The only exception was 2023-24, when strong winter peaks in northern states reduced the strength and significance of the association (p = 0.085). In both 2022-23 and 2024-25, latitude remained a strong, significant predictor (p <<< 0.001). These analyses demonstrate that the primary EI–latitude relationship is not an artifact of multi-year averaging.

(NEW) Supplemental Figure 2. Epidemic intensity calculated for each state and COVID-year in relation to latitude.

C. The paper summarizes timing with a single circular mean across years, but the distribution is explicitly biannual (summer and winter peaks). A single mean is not a sufficient statistic and can be misleading (means can land in troughs). Please report additional timing metrics: (a) circular dispersion, (b) separate peak weeks (summer/winter) per state/year, and (c) peak asymmetry (summer minus winter magnitude). Consider re-running the latitude regressions on these peak-specific outcomes.

We agree that the biannual structure of COVID-19 activity complicates interpretation of the mean case timing, which is typically most informative for pathogens with a single annual peak. Even so, we believe the mean case timing remains useful for indicating which peak - summer or winter - contributes more heavily to the annual distribution. States with more pronounced summer peaks tend to exhibit lower mean timing values, reflecting a greater concentration of cases during summer months. To contextualize this metric, we added an analysis of the winter-to-summer peak magnitude ratio for each COVID-year (NEW Supplemental Figure 2). Higher-latitude states exhibited larger winter-to-summer ratios, illustrating how relative peak strength varies geographically and across years. Together, these peak-ratio results complement and contextualize the mean case timing measure by illustrating how the relative strength of the two peaks shifts over time and across states.

(NEW) Supplemental Figure 3. Summer/Winter peak ratio for each state and COVID-year, where larger ratios indicate larger winter peaks relative to summer peaks.

D. Equation (3) includes state, year, and week fixed effects—a good start—but more detail is needed. Please:

Clarify whether two-way or three-way FE are included simultaneously (state FE + year FE + week-of-year FE). If week FE is common across years, it absorbs seasonality; then the climate effect is identified through deviations from the seasonal mean. Make this explicit and discuss implications.

Justify bin choices (5°C, 3 g/kg) analytically (e.g., using information criteria or cross-validation) rather than aligning reference bins to visual troughs (risk of post-hoc selection).

Show partial dependence plots with CIs and within-R² to emphasize within-state identification.

Thank you for the opportunity to clarify the fixed effects models. Yes, fixed effects are included simultaneously such that the effects of temperature and specific humidity are identified through deviations from the seasonal mean. This is now clarified in the text.

To assess whether results were sensitive to binning decisions, we repeated the fixed-effects analyses across nine alternative binning strategies (temperature bins of 2°C, 3°C, 4°C, and 5°C; specific humidity bins of 2, 3, and 4 g/kg). Across all specifications, AIC, BIC, and cross-validated RMSE varied by less than 4%, and the estimated effects of temperature and specific humidity remained directionally and substantively unchanged (NEW Supplemental Table 2). These results indicate that our chosen 5°C and 2 g/kg bins provide an appropriate balance of interpretability and statistical stability. Temperature followed an approximately normal distribution, while specific humidity exhibited a slight right-hand skew.

(NEW) Supplemental table 2: Sensitivity analyses to determine optimal temperature and specific humidity bin sizes.

Notably, we updated Figure 3a in the manuscript to include the updated bin sizes for specific humidity.

(UPDATED) Figure 3. Results from a binned fixed effects model assessing logged ED visits with a COVID-19 diagnosis. A. Results from the binned fixed effects model, showing a U-shaped relationship between specific humidity and logged ED visits, and a more linear relationship between temperature and logged ED visits. B. Surface plot indicating periods of predicted heightened COVID-19 activity from binned fixed effects model, where state-specific intercepts account for state-specific factors such as population density.

We included an updated partial dependence plot from the GAM to explore the marginal estimated effect of specific humidity on logged ED visits with at COVID-19 diagnosis, which is updated Supplemental Figure 3a (plot included in comment E response).

E. For the GAM [Equation (4)], please specify the exact basis, knot placement, smoothing parameter selection (REML is noted), and present edf, χ²/F-tests, and residual diagnostics (ACF of residuals, concurvity).

We appreciate the opportunity to elaborate and have now updated the GAM description to include more information. In the revised manuscript, we expanded the GAM description to specify spline bases, knot placement, smoothing parameter estimation (REML), and all inferential statistics. To address temporal autocorrelation, we added a cyclic cubic spline for epidemiologic week (s(week, bs=“cc”, k=20)), which enforces continuity across years. State fixed effects were included via factor(state). We now report edf, F-statistics, p-values, and concurvity metrics in NEW Supplemental Table 3 and provide full diagnostic plots in NEW Supplemental Figure 4. Diagnostics indicate approximate normality, minimal autocorrelation, and no major violations of model assumptions. We also added partial-effect plots for humidity and week (UPDATED Supplemental Figure 5).

(NEW) SI Table 3. GAM fit statistics.

Residual diagnostic plots, including residual vs. fitted, Q-Q plots, and the residual distribution, are included in (NEW) Supplemental Figure 4, confirming no remaining strong autocorrelation or systematic deviations from model assumptions. A Q-Q plot of deviance residuals (Supplemental Figure 4A) showed approximate normality with only minor tail deviations, indicating no major violations of distributional assumptions. Residuals had a normal distribution (Supplemental Figure 4B), and when plotted against the linear predictor (Supplemental Figure 4C) showed no systematic pattern or heteroskedasticity, indicating adequate mean-variance fit of the GAM. The response-fitted plot showed a positive linear association without systematic curvature (Supplemental Figure 4D), though with some residual variability, indicating that while the GAM captures broad mean patterns, climate factors alone explain only a modest share of week-to-week variation in ED visits with a COVID-19 diagnosis.

(NEW) Supplemental Figure 4. SI Diagnostic plots for the generalized additive model (Equation 4). A. Q-Q plot showing residuals largely following the theoretical normal distribution. B. Histogram of residuals, approximately symmetric and centered near zero. C. Residuals versus linear predictor, showing no major nonlinear patterns, with mild heteroskedasticity. D. Observed versus fitted values, indicating that fitted values span a narrower range than the observed log(ED visits). Together, these diagnostics support the adequacy of the model while motivating inclusion of state and cyclic week effects.

To interpret the shape and influence of the humidity spline, we present partial‐effect plots for the specific humidity and week-of-the-year spline terms for the generalized additive model (UPDATED Supplemental Figure 5).

(UPDATED) Supplemental Figure 5. Estimated smooth terms from the GAM including state and cyclic week effects. A. Specific humidity shows a nonlinear, U-shaped association with log(ED visits). B. The cyclic spline for week captures within-year seasonal structure. Solid lines represent fitted effects; dashed lines show 95% confidence intervals.

F. Weekly panel data will exhibit strong serial correlation. Clustering by state is necessary but not sufficient if residuals are highly persistent. Please report tests/plots of residual autocorrelation and repeat regressions with Driscoll–Kraay or Newey–West (panel) robust errors (or at least state-level AR(1) correction) to verify inference stability.

Thank you for bringing this to our attention, and we agree that weekly panel data can exhibit serial correlation. To address this, we re-estimated our model using Newey-West heteroskedasticity and autocorrelation (HAC) standard errors. We conducted a joint Wald test of the spline basis coefficients for both s(week_q) and s(week), and in both cases, the smooth terms remained statistically significant after the HAC correction, indicating that our primary inference remained robust to serial correlation. We report these HAC-adjusted tests with the GAM fit statistics in (NEW) Supplemental Table 3. Importantly, the initial conclusions of the GAM, being the strong nonlinear association between specific humidity and a nonsignificant linear effect of temperature, remain consistent under the HAC-robust analysis.

(NEW) SI Table 3. GAM fit statistics.

G. Neighboring states share weather and mobility catchments; ignoring spatial autocorrelation can bias SEs and inflate significance. Perform Moran’s I on residuals and, if material, implement spatial HAC corrections or include region×week FE. Alternatively, estimate a spatial error model on state-level annual summaries as a sensitivity analysis.

We appreciate the opportunity to look into potential spatial autocorrelation. We evaluated spatial autocorrelation in the state-level mean residuals from the GAM using global Moran’s I with border-or-corner adjacency (queen contiguity) weights (with 999 permutations). Moran’s I was -0.137 with a permutation p-value of 0.874, indicating no evidence of spatial clustering. This suggests that residual spatial dependence is minimal and does not substantially affect the findings. These results, combined with the inclusion of both state and week fixed effects and the Newey-West robust standard errors, indicate that our findings are stable to serial and spatial correlation.

H. Summer/winter peaks in ED COVID percentages may be entangled with RSV/flu/enterovirus waves, school terms, AC/indoor time, wildfire smoke, or holiday periods. At a minimum, add controls for:

Influenza and RSV ED indicators (or ILI/SARI proxies) by state-week.

Mobility or policy stringency (the Discussion notes this as future work; it belongs in robustness today).

School in/out of session indicators.

Wildfire smoke (PM2.5) has affected recent summers, particularly those with smoke-affected conditions. Include these in sensitivity models to demonstrate that climate associations are not proxies for u

---

## [Decision Letter · Decision Letter 1]

26 Jan 2026

Spatial patterns and environmental influences of COVID-19 outbreaks, post-Omicron

PONE-D-25-57696R1

Dear Dr. Stamper,

We’re pleased to inform you that your manuscript has been judged scientifically suitable for publication and will be formally accepted for publication once it meets all outstanding technical requirements.

Kind regards,

Yury E Khudyakov, PhD

Academic Editor

PLOS One

Additional Editor Comments (optional):

Reviewers' comments:

Reviewer's Responses to Questions

**Comments to the Author**

1. If the authors have adequately addressed your comments raised in a previous round of review and you feel that this manuscript is now acceptable for publication, you may indicate that here to bypass the “Comments to the Author” section, enter your conflict of interest statement in the “Confidential to Editor” section, and submit your "Accept" recommendation.

Reviewer #1: All comments have been addressed

2. Is the manuscript technically sound, and do the data support the conclusions?

Reviewer #1: Yes

3. Has the statistical analysis been performed appropriately and rigorously?

Reviewer #1: I Don't Know

4. Have the authors made all data underlying the findings in their manuscript fully available?

Reviewer #1: Yes

5. Is the manuscript presented in an intelligible fashion and written in standard English?

Reviewer #1: Yes

6. Review Comments to the Author

Reviewer #1: The authors have appropriately addressed all the concerns raised in the previous review.The manuscript has been improved and is now suitable for publication

7. PLOS authors have the option to publish the peer review history of their article (what does this mean?). If published, this will include your full peer review and any attached files.

Reviewer #1: No

---

## [Editor Report · Acceptance letter]

PONE-D-25-57696R1

PLOS One

Dear Dr. Stamper,

I'm pleased to inform you that your manuscript has been deemed suitable for publication in PLOS One. Congratulations! Your manuscript is now being handed over to our production team.

Kind regards,

on behalf of

Dr. Yury E Khudyakov

Academic Editor

PLOS One